# Effects of 12-Month Training Intervention on Physical Fitness, Body Composition, and Health Markers in Finnish Navy Soldiers

**DOI:** 10.3390/healthcare11192698

**Published:** 2023-10-09

**Authors:** Mikko Myllylä, Kai I. Parkkola, Tommi Ojanen, Olli J. Heinonen, Juha-Petri Ruohola, Tero Vahlberg, Heikki Kyröläinen

**Affiliations:** 1Centre for Military Medicine, The Finnish Defence Forces, 20241 Turku, Finland; 2Doctoral Programme in Clinical Research, University of Turku, 20014 Turku, Finland; 3Faculty of Medicine and Health Technology, Tampere University, 33100 Tampere, Finland; 4Department of Leadership and Military Pedagogy, National Defence University, 00861 Helsinki, Finland; 5Human Performance Division, Finnish Defence Research Agency, The Finnish Defence Forces, 04310 Tuusula, Finland; 6Paavo Nurmi Centre & Unit for Health and Physical Activity, University of Turku, 20520 Turku, Finland; 7Defence Command Finland, The Finnish Defence Forces, 00131 Helsinki, Finland; 8Department of Biostatistics, University of Turku, 20014 Turku, Finland; 9Neuromuscular Research Center, Faculty of Sport and Health Sciences, University of Jyväskylä, 40014 Jyväskylä, Finland

**Keywords:** exercise, intervention study, training program, voluntary program, long-term effect, workplace, military personnel, navy personnel

## Abstract

Background: Most Western adults do not meet the recommendations for sufficient activity, and obesity is a global problem. Similar trends are also seen among Western military personnel. Many successful physical training interventions have been carried out in military environments, but the interventions have been quite short term, and the training has been supervised. Therefore, the aim of this study was to investigate the effects of a 12-month voluntary motivational training intervention among the Finnish Defence Forces’ (FDF) Navy soldiers. Methods: In total, 77 FDF Navy soldiers, serving in missile patrol boats, took part in the study. The intervention group (IG) contained 45 participants and the control group (CG) contained 32 participants. The IG was divided into four teams that carried out the intervention, while the CG took part in only the measurements. Results: Most of the participants (65%) in the IG reported that they had increased their exercise volume during the intervention, but no major beneficial impacts on the physical fitness, body composition, or health markers were observed. Nevertheless, there was a clear diversity visible between the subgroups in the IG. The team that reported the most exercise had the best motivation and the most motivated team coach and also had the most improved physical fitness and body composition results. Conclusions: The present study points out that in military environments, long-term voluntary training interventions may not be as successful as short-term supervised interventions. The results also suggest that in voluntary training interventions among military personnel, the participants’ motivation to exercise is a key factor when improving physical fitness.

## 1. Introduction

Physical inactivity and obesity are risk factors for many health conditions, e.g., type II diabetes, cardiovascular diseases, and certain cancers [1,2]. Nevertheless, most Western adults do not meet the physical activity recommendations [1], and since 1975, obesity has nearly tripled worldwide [2]. Especially in Western countries, men’s endurance fitness at the beginning of military service has also decreased, and, simultaneously, their mean body mass has increased [3,4]. A study in the U.S. pointed out that considering military personnel aged 20 years or older, there were more obese or overweight people in the navy (67.0%) than in any the other military branch [5]. In the naval environment, obesity is considered to increase danger on-board because it may be challenging for obese people to complete physically demanding duties, e.g., during an emergency [6]. Obesity may also induce problems since it is a risk factor for cardiovascular diseases and diabetes, which may cause problems if they emerge on-board [2,6].

Both strength and endurance capacity are recognized as important performance factors among soldiers when considering military readiness [7,8,9]. In the maritime environment, the personnel’s habits of maintain regular physical activity, good physical fitness, an athletic body composition, and a healthy lifestyle may all be beneficial factors for preventing sleepiness, fatigue, and stress responses on-board [10]. Regarding endurance capacity, it has been reported that naval personnel should have a maximal oxygen uptake (VO_2_max) of 41 mL/kg/min as a minimum standard in case of firefighting on-board [11]. In addition to firefighting on-board, the following tasks are considered to be physically demanding: casualty handling, damage control, and basic transition duties [12]. Out of these tasks, casualty handling is considered to be the most challenging naval duty regarding strength and endurance capacity. Maximal strength (e.g., grip strength), muscular endurance, the absolute VO_2_max, and anaerobic capacity have been reported as essential factors when predicting the performance in casualty evacuation [13,14,15,16].

The working environment has been promoted as an ideal area for health promotion and physical activity interventions because it offers the possibility to contact large diverse groups and subjects that might otherwise not be physically active [17,18]. However, a systematic review showed that 7/12 randomized, controlled physical activity interventions at the workplace had no effect [19]. Another systematic review suggested that although some interventions may have beneficial effects, their overall effect is inconclusive [20]. Recent research suggests that combined strength and endurance training is the most optimal training method to enhance soldiers’ overall physical performance [21]. Regarding naval personnel, a prior study investigated the effects of an 8-week linear periodized strength training program for Naval Academy cadets in addition to their daily training [22]. The additional strength training of the experimental group consisted of exercises such as bench presses, squats, lunges, arm curls, etc. In 8 weeks, the task-specific muscle strength, power, and agility of the experimental group increased when compared to the control group. Among military personnel, many other successful interventions with different training programs (strength, endurance, combined strength, and endurance training) have also been carried out [21,23]. However, the interventions have been quite short term (usually 5–12 weeks) and the training has been supervised [21]. A meta-analysis investigating fitness outcomes from motivational physical activity interventions suggested that supervised interventions may need trained personnel, which can add costs to the interventions, where again motivational interventions do not necessarily need as much specific equipment or personnel [24]. It also pointed out that in the civilian environment, motivational interventions, designed to increase physical activity, may improve endurance capacity in healthy adults.

The aim of this study was to investigate the effects of a 12-month voluntary motivational training intervention among the Finnish Defence Forces’ (FDF) Navy soldiers. The intervention included a training program that contained two primary and two secondary weekly training sessions (1 h each). The intervention was performed utilizing the personnel’s possibility to use two hours of their working time for physical training weekly. It was hypothesized that by carrying out the voluntary intervention, it would be possible to increase physical activity and improve the muscle strength, endurance, and health markers of the navy soldiers.

## 2. Materials and Methods

### 2.1. Study Design and Study Participants

In total, 77 FDF Navy soldiers serving in missile patrol boats took part in the study. The intervention group (IG) contained 45 participants, and the control group (CG) consisted of 32 participants. The participants in the IG and CG served in two separate missile patrol boat fleets. The IG was further divided to four teams (T1–4) so that each team comprised participants from the same vessel. The outline of the study design is shown in Figure 1. Two study participants in the CG were female; all the other participants were male. The demographics of the participants are shown in Table 1.

The IG carried out a 12-month training intervention, while the CG took part in only the measurements of the study. All members from the studied fleets that were considered workwise to be able to take part in the measurements during the study period were recruited for the study. Those fleet members who at the beginning of the study were known to deploy elsewhere during the study period were not recruited. The participants gave their informed consent for voluntary participation and did not gain any financial advantage from the study.

The study measurements were performed at baseline (PRE) and after the 12-month study period (POST). Additionally, 6-month (MID) measurements were performed in the middle of the study period for the IG. The timeline of the study measurements is shown in Figure 2.

#### 2.1.1. Intervention Group (IG)

The following steps were performed for the IG:Once the IG was divided into four teams, each team nominated its own coach and a back-up coach from the team members. The coaches gave instructions, organized training sessions, and verbally motivated their team members to carry out the weekly training.On a weekly basis, the participants electronically reported on the completion of the training program and also all other daily exercises in minutes. The main researcher (MM) motivated the teams by sending a weekly e-mail and delivered each week’s training program. During the 12-month study period, a weekly e-mail served as a constant reminder that the intervention was going on. In the e-mail, depending on the reported volume of weekly training, the MM either thanked the IG for a good work or encouraged them to add training volume. At the beginning of the intervention, the MM also gave a 2 h lecture separately to each team about the importance of proper nutrition and physical activity regarding health.To further enhance the motivation of the IG, the group was informed that two teams would be rewarded at the end of the intervention. The rewarding was primarily based on the increase in physical fitness test results during the intervention. The self-reported following of the training program and reported amounts of other physical activity were also taken into consideration. The rewarding was based on scoring so that the two teams that collected the most points would be rewarded. The reward was team participation in a chosen sport during one working day. The team that scored the most points was also rewarded with an award plaque. The details of the scoring are shown in Appendix A.

#### 2.1.2. Control Group (CG)

The CG took part only in the PRE and POST measurements. No intervention, other information, or motivation were given.

### 2.2. Training Program

The training program included strength training, endurance training, and combined strength and endurance training. The program consisted of two weekly primary training sessions (1 h each) and secondary training sessions (1 h each). The primary training sessions were performed utilizing the personnel’s possibility to use two hours of their working time for physical activity weekly. The two secondary training sessions were to be performed during the participants’ free time. Every fourth week was physically lighter, containing only the two primary training sessions. The weekly primary training sessions were advised to be completed on Monday and Thursday, and the secondary training sessions on Tuesday and Friday. During weeks on duty at sea, the participants were advised to perform only two primary training sessions containing strength training especially designed for the service at sea. During the study period, the participants were on duty at sea for 90–110 days in total. More details of the training program are shown in Appendix A.

### 2.3. Measurements

Primary physical fitness measures included the number of push-ups and sit-ups in one minute, a standing long jump, and a 12 min run test. Push-ups and sit-ups measured the dynamic muscle endurance of the upper extremities, abdominal muscles, and hip flexors [25,26]. The standing long jump assessed the power production of the lower extremities [27]. A 12 min run test estimated the aerobic capacity [28]. All participants were accustomed to these primary physical fitness tests because they are included in basic military training.

In addition, the maximal voluntary contraction of the lower (MVClower) and upper (MVCupper) extremities were assessed with electromechanical dynamometers manufactured by the University of Jyväskylä (Jyväskylä, Finland). The measurements were performed bilaterally in a sitting position. For the MVClower, the knee and hip angles were maintained at 107° and 110° [29], respectively. For the MVCupper, a handlebar was set at the height of the participants’ shoulders and maintained a 90° angle with the elbows. A seated medicine ball throw (SMBT) was conducted to evaluate the participants’ explosive power production of the upper extremities [30]. It was performed on the floor in a sitting position with the legs fully extended and the back kept against a wall. The medicine ball (weight 2 kg) was held with both hands and the forearms were maintained parallel to the floor. The medicine ball was thrown ahead with maximal effort and the distance from the wall to the landing position was measured. A supervisor demonstrated the correct techniques for the fitness tests and all the tests were supervised.

The body mass, skeletal muscle mass, fat mass, and body fat percentage were measured using a segmental multifrequency bioimpedance analysis in the morning after a 10-h fast. (InBody 720, Biospace, Seoul, Republic of Korea). The participants’ height was measured and the body mass index (BMI) was calculated. The waist circumference was measured according to WHO guidelines [31].

The resting blood pressure was measured in a sitting position from the nondominant hand according to WHO guidelines [31]. The resting heart rate was recorded during the blood pressure measurements, and the mean of three separated measurements was used. Venous blood samples were collected in the morning after a 10 h fast from the antecubital vein. The samples were centrifuged (Labofuge 300, Heraeus Instruments GmbH, Hanau, Germany) at 2200× *g* for 15 min, and plasma was separated and kept at 4 °C prior to analysis. Whole-blood samples were kept at room temperature. All the samples were directly analyzed at TYKS Laboratories at Turku University Central Hospital. The analyses included plasma glucose (FPG) (photometry), blood hemoglobin A1c (HbA1c) (immunoturbidimetry/Roche/Cobas 6000 c 501), fasting plasma insulin (FPI) (ECLIA/Roche/cobas 8000 e 801), total cholesterol (TC) (enzymatic (CHOD-PAP)/Roche/Cobas 8000 c 702), low-density lipoprotein cholesterol (LDL-C) (enzymatic (direct)/Roche/Cobas 8000 c 702), high-density lipoprotein cholesterol (HDL-C) (enzymatic (direct)/Roche/Cobas 8000 c 702), and triglycerides (TG) (enzymatic (GPO-PAP)/Roche/Cobas 8000 c 702).

At the end of the intervention, participants from the IG answered a questionnaire and reported if they increased their exercise volume during the study period due to the intervention when compared to the preceding 12 months (1 = yes, 2 = no). The team coaches reported their own subjective motivation toward their duty as team coach and the coaches’ views of the overall motivation of the team toward the intervention on a scale. The reported scale consisted of numeric values: 1 (very low motivation), 2 (low motivation), 3 (not low or high motivation), 4 (high motivation), and 5 (very high motivation). The team coaches also reported their views on the overall difficulty to motivate their team to exercise regarding the intervention on a scale. The scale consisted of numeric values: 1 (very easy), 2 (easy), 3 (not difficult or easy), 4 (difficult), and 5 (very difficult).

### 2.4. Statistics

A statistical analysis was performed using the SPSS statistical software (SPSS version 27.0.1.0; SPSS Inc., Chicago, IL, USA). Normal distribution of the data was assessed by skewness (values between −1 and 1 were considered as normally distributed) and visually by histogram. Prior to the statistical analysis, log- and 1/x-transformations were used for non-normally distributed variables. For non-normally distributed variables for which no appropriate transformations were found, a nonparametric test was used. Comparisons of normally distributed variables between the CG and IG and between the CG and teams were performed using the independent samples *t*-test. For non-normally distributed values, a Mann–Whitney U test was used. A paired samples *t*-test was used to evaluate the difference between normally distributed PRE and POST values in the CG. A linear mixed model was used to evaluate the mean changes between normally distributed and transformed PRE, MID, and POST values in the IG and teams to account for the repeated measurements of the participants. In the linear mixed model, a Bonferroni correction was used when the MID and POST values were compared to the PRE values. For non-normally distributed values, a Wilcoxon signed-rank test was used to compare repeated measurements. Effect size for the difference between IG and CG for PRE–POST mean change was calculated with Cohen’s D. In the correlation analysis, a Spearman correlation coefficient was used. *p*-values lower than 0.05 were considered significant.

## 3. Results

### 3.1. Physical Fitness Tests

There were no major differences over time in the physical fitness test results in the IG, CG, or between the groups. Only the mean change for the MVClower result was higher in IG when compared to the CG (13 vs. −23 kg, *p* = 0.013).

On a team level, when compared over time to the CG, T4’s results increased for sit-ups (3 vs. −2 reps/min, *p* = 0.005) and T2’s results decreased for the standing long jump (−7 vs. −1 cm, *p* = 0.018). Within the teams, T4 also increased their results for sit-ups (3 reps/min, *p* = 0.006) and the standing long jump (3 cm, *p* = 0.044). T2’s results decreased for the standing long jump (−7 cm, *p* = 0.012) and at the 6-month measurement point their results also decreased for SMBT (−15 cm, *p* = 0.034). All physical fitness test results are shown in Appendix A.

### 3.2. Body Composition

In the IG, the body mass increased over time when compared to the CG (1.3 vs. −0.4 kg, *p* = 0.015). Within the IG, the body mass also increased (1.3 kg, *p* < 0.001), and the increase was visible already at the 6-month measurement point (0.9 kg, *p* = 0.024). However, there were no significant differences in body fat percentage, skeletal muscle mass, or waist circumference between the groups over time.

On a team level, when compared over time to the CG, T2 had an increase in body mass (2.8 vs. −0.4 kg, *p* < 0.001) and fat mass (2.3 vs. −0.3 kg, *p* = 0.008). Within the teams, in T2 the body mass and fat mass also increased (body mass 2.8 kg, *p* < 0.001; fat mass 2.3 kg, *p* = 0.014), and the increase was already seen at the 6-month measurement point (body mass 2.2 kg, *p* < 0.001; fat mass 1.5 kg, *p* = 0.049). Within T2, there was also an increase in the body fat percentage at the 12-month measurement point (1.9%, *p* = 0.029). Within T4, the skeletal muscle mass increased at the 6-month measurement point (0.7 kg, *p* = 0.024), but no change was seen at the 12-month measurement point. The results denote that the intervention had only a weight-gaining effect, and over time, the body composition results of the teams were very variable. The changes in the participants’ body composition during the study period are shown in Figure 3.

### 3.3. Blood Pressure, Heart Rate, and Blood Biomarkers

The resting blood pressure and heart rate remained similar over time in the IG and CG. For the blood biomarkers, TC decreased more in the CG than in the IG (−0.4 vs. −0.2 mmol/L, *p* = 0.049). Also, TG increased in the IG when compared to the CG (0.2 vs. −0.1 mmol/L, *p* = 0.012). On the other hand, there was a decrease in the HDL-C values within the CG (−0.1 mmol/L, *p* = 0.003). Within the IG, the HDL-C values had increased at the 6-month measurement point (0.1 mmol/L, *p* = 0.031), but there was no statistically significant difference at the 12-month measurement point. Within both the IG and CG, there was a decrease in the LDL-C values (IG −0.2 mmol/L, *p* = 0.021; CG −0.2 mmol/L, *p* = 0.026). Within the teams, in T2, the TC and LDL-C values decreased (TC −0.5 mmol/L, *p* = 0.002; LDL-C −0.5 mmol/L, *p* = 0.004), and the decrease was visible already in the 6-month measurements (TC −0.5 mmol/L, *p* = 0.005; LDL-C −0.5 mmol/L, *p* = 0.002). Overall, these results denote that the intervention did not have major effects on cholesterol concentrations.

Glucose metabolism (HbA1c, FPG, FPI) remained similar over time in both IG and CG. Within both the IG and CG, the HbA1c values increased (IG 1.5 mmol/mol, *p* < 0.001; CG 1.2 mmol/mol, *p* < 0.001) and the increase was also visible within all teams in the IG. Within the CG, the FPI increased (1.5 mU/L, *p* = 0.019) at the 12-month measurement point, and within the IG, the FPG had increased in the 6-month measurements (0.1 mmol/L, *p* = 0.029). These results for FPG, HbA1c, and FPI denote that the intervention did not have major effects on glucose metabolism. The changes in the participants’ FPG, HbA1c, TC, LDL-C, HDL-C, and TG values during the study period are shown in Figure 4.

### 3.4. Self-Reported Weekly Exercise Volume

In the IG, the mean volume (±SD) of weekly exercise during the intervention was 73 ± 53 min in total, including 42 ± 35 min of endurance training and 30 ± 25 min of strength training. T4 reported the largest volumes of weekly exercise, on average 124 ± 44 min in total, including 64 ± 34 min of endurance training and 60 ± 20 min of strength training. In the IG, the mean percentage of the training program completed was only 8%. For T4 the training program was carried out the most (12%) and for T2 the least (2%). The reported weekly exercise volume during the intervention is shown in Table 2.

### 3.5. The Reported Subjective Effect of the Intervention

In the IG, 65% of the participants reported they performed more exercises during the study due to the intervention than before, while 35% reported that the intervention had no effect. In T4, most participants (83%) reported increased exercise volume during the study period, and in T2, most participants (60%) reported no intervention effect. The reported subjective effect of the intervention to the exercise volume during the study period on a personal level is shown in Figure 5.

### 3.6. The Reported Opinions from the Team Coaches

The coach for T4 reported the highest value (5/5) on the numeric scale for motivation toward duties as team coach. The coach also reported the highest value (5/5) for the motivation of the team toward the intervention and reported that it was very easy (1/5) to motivate team members to exercise regarding the intervention. For T2, the reported motivation of the coach was 3/5 on the numeric scale. The coach reported the lowest value (2/5) for the motivation of the team and also reported that it was difficult (4/5) to motivate team members to exercise regarding the intervention. In general, the mean of the reported opinions of all team coaches revealed that the own motivation of the coaches themselves toward their duties was rather high (mean 3.6/5), they reported that the motivation of the team members toward the intervention was not high or low (mean 3.1/5) and they reported that the difficulty to motivate team members to exercise regarding the intervention was not difficult or easy (mean 3.3/5).

### 3.7. The Relation between Data and Self-Reports in IG

Between the PRE and POST measurements, the participants who reported having increased exercise volume due to the intervention had increased results for sit-ups (2 ± 4 reps/min, *p* = 0.010), push-ups (3 ± 5 reps/min, *p* = 0.043), and standing long jump (14 ± 5 cm, *p* = 0.004). The reported total exercise volume was positively associated with sit-ups (r = 0.48, *p* = 0.002), the standing long jump (r = 0.52, *p* = 0.001), and 12 min run test (r = 0.40, *p* = 0.037) and negatively associated with waist circumference (r = −0.33, *p* = 0.046). The reported endurance training time was also positively associated with the results for the 12 min run test (r = 0.38, *p* = 0.048), sit-ups (r = 0.49, *p* = 0.002), and the standing long jump (r = 0.41, *p* = 0.010) and negatively associated with waist circumference (r = −0.38, *p* = 0.018). The reported strength training volume was positively associated with the results for sit-ups (r = 0.43, *p* = 0.007) and the standing long jump (r = 0.57, *p* < 0.001). Following the training program was positively associated with the results for the standing long jump (r = 0.39, *p* = 0.015) and negatively associated with waist circumference (r = −0.36, *p* = 0.028), but it was also negatively associated with skeletal muscle mass (r = −0.36, *p* = 0.041).

In the PRE–POST measurement results, the motivation of the team coach was positively associated with sit-ups (r = 0.51, *p* = 0.001). In the coach’s view, the motivation of the team members was also positively associated with the results for sit-ups (r = 0.33, *p* = 0.042) and the standing long jump (r = 0.53, *p* = 0.001) and negatively associated with body mass (r = −0.45, *p* = 0.005), fat mass (r = −0.42, *p* = 0.008), body fat percentage (r = −0.41, *p* = 0.009), and waist circumference (r = −0.44, *p* = 0.031). Increased difficulty to motivate the team members to exercise regarding the intervention was associated with decreased results for sit-ups (r = −0.42, *p* = 0.007) and the standing long jump (r = −0.33, *p* = 0.041). The results denote that a higher reported training volume and motivation of the coaches/team members had mainly positive associations with the physical fitness test results and mainly positive influence on the body composition parameters.

## 4. Discussion

The aim of this study was to investigate the effects of a 12-month voluntary intervention among navy soldiers. It was hypothesized that by carrying out the intervention, it would be possible to increase physical activity and thus improve the muscular strength, endurance, and health markers of the participants.

In the IG, most of the study participants (65%) reported that because of the intervention, they exercised more during the study period than the 12 months before it. This finding is in line with a systematic literature review about workplace physical activity programs, which points out that interventions in the workplace may increase physical activity [32]. Still, the mean of the reported weekly exercise in the present study was 73 min, which was a little bit over a half of the personnel’s weekly possibility to exercise during working time. There was a clear diversity visible between the teams considering the reported outcomes. T4 reported the most weekly exercise (124 min), which was 4 min over the personnel’s weekly possibility to exercise during working time. From T4, most participants (83%) also reported that they had added exercise volume because of the intervention during the study period. The coach for T4 reported very high motivation, reported that the members of the team had also had very high motivation and reported that it was very easy to motivate the team members to exercise regarding the intervention. This is in line with the current literature, where motivation is considered as a key factor in terms of physical activity, and it is also considered an important component in exercise adherence [33,34,35].

In physical fitness tests, despite the intervention, there was a difference between IG and CG only in the results for the MVClower. This indicates that considering the physical fitness test results, even though most of the participants had added to their exercise volume because of the intervention, it was not enough to make a general beneficial impact on the physical performance of the study participants. This finding is in line with previous studies, where it has been found that although the workplace physical activity interventions can increase physical activity [32], the overall findings considering interventions are quite inconclusive [20]. In the present study, it is logical that the reported exercise volume (only 73 min as a weekly mean) in the IG did not lead to improvements in physical fitness tests. On the other hand, naval soldiers have a strict selection process and requirements for the minimum level of physical fitness, and soldiers overall have been found to be more physically active than civilians [36]. Therefore, it may not always be strictly necessary to enhance soldiers’ physical fitness if they have maintained it at a satisfactory level.

For the military, including naval personnel, successful studies have been reported which have enhanced the physical performance of the participants by using different training interventions and programs [21,22,23]. Still, most of the studied training interventions have been performed in a supervised manner and the interventions have been quite short term, usually lasting 5–12 weeks. Nevertheless, in a crisis management environment and among special operation forces, there have been reported successful 6-month enduring training interventions [37,38]. A study investigating two different training programs for conscripts, experienced a plateau in the increase in strength and power measurements of the participants when the volume of controlled training decreased while the volume of field training increased [39]. The study suggested that with structured, regular, and controlled physical training, it is possible to achieve better results considering conscripts’ physical performance than otherwise. This and the present study results point out that in a military environment, carrying out a training intervention in a supervised manner may be more important than in a civilian environment, where it has been found that motivational physical activity interventions alone may have the potential to improve physical performance in healthy adults [24]. This may be because the military is a controlled environment and works via orders, which could potentially inhibit the completion of uncontrolled voluntary duties. Nevertheless, for military employees, there is an upcoming study on the use of an accelerometer smartphone application to increase employees’ physical activity, which might shed more light on the effectiveness of motivational interventions in the military context [40]. The reporting of possible negative results is also important, because in civilian motivational physical activity interventions, there may be some publication bias considering the lack of small studies with negative findings [24].

The environment on a missile patrol boat is quite confined, and although strength training was possible on the deck of the vessel, endurance training had to be performed on land. This might impair the endurance performance of the naval soldiers who work on small vessels and do not have the possibility to exercise on land during duty at sea. In the military context, this issue concerns mainly naval soldiers, while the same kind of long-lasting work in confined environment does not occur in other military branches. In the present study, there were no significant changes considering 12 min run test results, but the outbreak of COVID-19 prohibited the conduction of the test to many participants at the POST-measurements, making the assessment of changes in the participants’ endurance fitness incomplete.

On a team level, there was a clear diversity between the teams in physical fitness tests, and the results were in line with the reported weekly exercise volumes and the views of the coaches. T4 that reported the most exercises and had the highest motivation regarding to the report of the team’s coach, also had the most beneficial results. T2, in which only 40% of the participants reported that they had increased the volume of exercises because of the intervention, had the lowest motivation reported by the team’s coach and reported much less exercises, also had the least beneficial results. This finding suggests that, in addition to the fact that motivation is a key factor considering physical activity [33,34,35], it is also an important component when the aim is to enhance physical performance.

There was an increase in body mass in the IG when compared to CG. Still, in the IG there were no other significant changes in body composition. In T2, there was an increase in body mass and fat mass within the team and also when compared to CG. However, within T4 skeletal muscle mass increased at the 6-month measurements. This points out that the intervention was, in general, not effective enough to create beneficial effects on body composition in the IG. The results between the teams were logical, while exercising is known to lead to a healthier body composition through the reduction of body fat [41], insufficient volume of exercise may result in the opposite results.

There were no differences in the changes of the resting blood pressure or heart rate between the IG and CG during the study. Considering cholesterol levels, the TC decreased more in the CG when compared to IG and TG increased in the IG when compared to CG. On the other hand, there was a decrease in the HDL-C values within the CG, which reduced the level of TC. On a team level, the results were quite inconclusive, while T2 that gained the most fat mass, had reductions in TC and LDL-C. This result is not in line with prior findings, while in soldiers, a higher amount of body fat has been linked with higher TC and LDL-C levels [42]. On the other hand, it is possible that the changes in the body composition of T2 were not great enough to have an effect on the TC and LDL-C. Considering the glucose metabolism (HbA1c, FPG, FPI), there were no differences between the IG and CG. Within both the IG and CG, there was still an increase in HbA1c. Within the IG there was also an increase in FPG and within the CG there was an increase in FPI. Exercising and a healthy diet lead to lower fat mass together with high muscle mass, which should reduce insulin resistance [43]. In the present study, considering the IG and CG, there was an increase in insulin resistance present as the results refer to an impaired glucose metabolism in both groups. In overall, the intervention was not effective enough to lead to significant beneficial health effects considering blood pressure, heart rate, or blood biomarkers. In the current research, occupational physical activity, in general, has been reported to both favorably and unfavorably associate with health-related outcomes and more research is needed to determine the health effects of occupational physical activity [44].

The strength of the present study is the good participation rate. A review study on health promotion interventions in workplaces found that interventions gathered in median only 33% participation rate [18], and the interventions usually attracted already fit participants [45]. This may affect the outcomes, while physical activity and health promotion interventions at workplaces are suggested to have four times more effective outcomes when the participation rates are low [46]. In the present study, all possible fleet members participated, and the results were obtained from 89% of the participants in the IG and from 84% in the CG at the 12-month measurements. The present study investigated a long-term voluntary training intervention in a military environment, which is an exception in the field of military training interventions, while usually the studied interventions have been quite short-term and the training has been performed in a supervised manner [21]. There has also been a lack of knowledge considering training interventions carried out with naval personnel. The present study provides deeper knowledge on long-term motivational voluntary training interventions in a naval environment.

A limitation of the present study is the considerably small number of study participants who worked on a certain vessel class. Another limitation of the present study is that we did not have information on the diets of the individual participants. During the study periods we also did not have any reports of the amount of physical activity of the participants in the CG, while valid weekly reports were considered too challenging to obtain without any kind of intervention or motivation. The outbreak of COVID-19 prohibited the conduction of the 12 min run test to many participants at the 12-month measurements, so the assessment of changes in endurance fitness of the participants during the study period were incomplete. On small vessels, endurance training may have to be performed on land, which may impair the endurance performance of the naval soldiers who work on small vessels and do not have the possibility to exercise on land during their duty at sea. The same does not go for larger naval ships, which are not that confined and enable the possibility to perform endurance training on board. The present study was designed to increase the physical activity of the naval soldiers in practice. Because the present study was not designed solely for research purposes, the data, e.g., the scoring of the teams, were quite complex. Considering further research, more studies with a larger number of participants from different ship classes are needed.

## 5. Conclusions

In the present study, most of the participants (65%) in the IG reported that they had increased their exercise volume because of the intervention during the study period. Still, the long-term (12-month) voluntary intervention for the navy soldiers did not have any major general beneficial impact on the physical fitness, body composition, or health markers of the study participants. This points out that in a military environment, even though there have been many successful short-term training interventions performed in a supervised manner reported [21], long-term voluntary motivational training interventions may not be as successful. In the present study results, there was clear diversity visible between the teams (T1–4) in the IG. T4, which reported the most exercise, had the best motivation, and had the most motivated team coach, also had the most beneficial results in the case of physical fitness and body composition. Motivation is considered a key factor in terms of physical activity, and it is also considered an important component in exercise adherence [33,34,35]. The present study results highlight the major importance of the participants’ motivation to exercise also when performing voluntary physical training interventions in a military environment.

## Figures and Tables

**Figure 1 healthcare-11-02698-f001:**
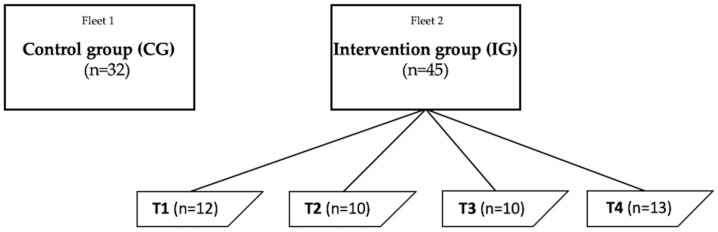
The outline of the study design. T1 = team 1, T2 = team 2, T3 = team 3, and T4 = team 4. Each team in the IG served in the same vessel.

**Figure 2 healthcare-11-02698-f002:**
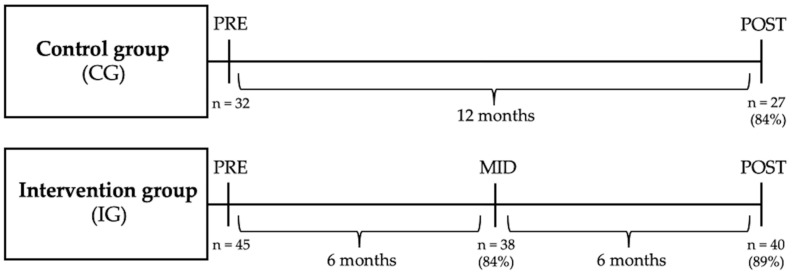
The timeline of the study measurements. PRE = baseline measurements, MID = measurements in the middle of the study period, POST = measurements after the 12-month study period. In the IG, two participants did not fulfill the intervention: one due to low motivation and the other because of an injury not related to the intervention. Three participants in the IG and five in the CG were deployed to other duties by the employer during the study and did not take part in the POST measurements.

**Figure 3 healthcare-11-02698-f003:**
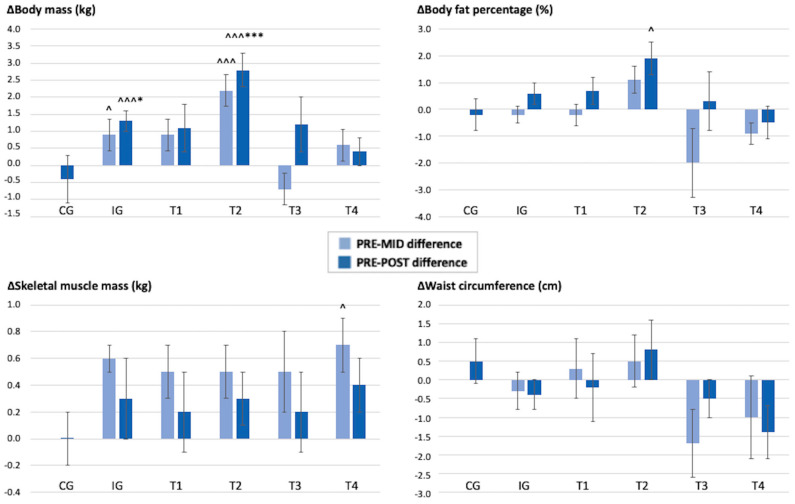
Mean (SD) changes in the participants’ body composition during the 12 months. PRE–MID difference = difference between the results at the baseline and at 6 months. PRE–POST difference = difference between the results at the baseline and at 12 months. CG = control group, IG = intervention group, T1 = team 1, T2 = team 2, T3 = team 3, T4 = team 4. * Significant difference compared to the control group, *p* < 0.05, *** *p* < 0.001. ^ Significant difference compared to the baseline value, *p* < 0.05, ^^^ *p* < 0.001.

**Figure 4 healthcare-11-02698-f004:**
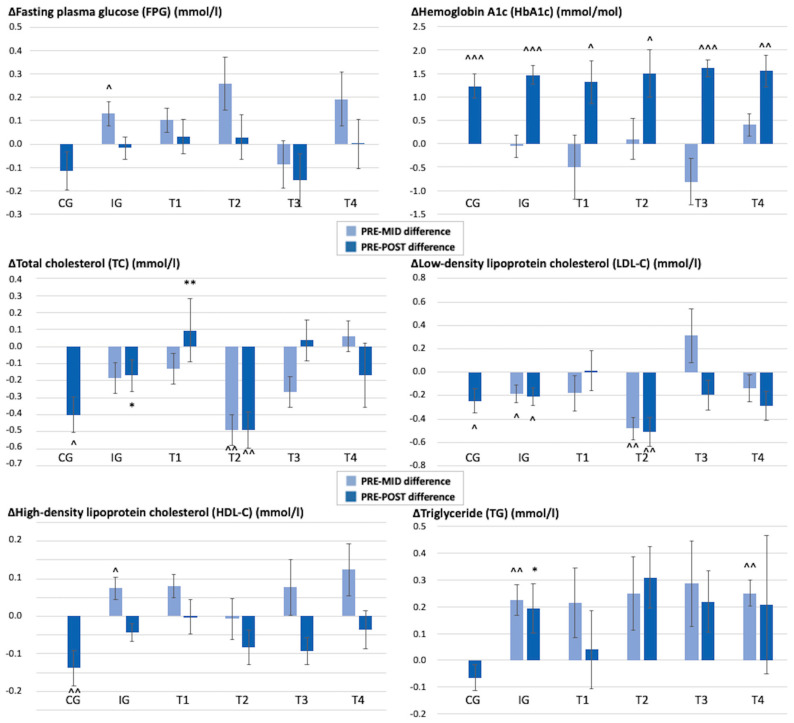
Mean (SD) changes in the participants’ FPG, HbA1c, TC, LDL-C, HDL-C, and TG values during the 12 months. PRE–MID difference = difference between the results at the baseline and at 6 months. PRE–POST difference = difference between the results at the baseline and at 12 months. CG = control group, IG = intervention group, T1 = team 1, T2 = team 2, T3 = team 3, T4 = team 4. * Significant difference compared to the control group, *p* < 0.05, ** *p* < 0.01. ^ Significant difference compared to the baseline value, *p* < 0.05, ^^ *p* < 0.01, ^^^ *p* < 0.001.

**Figure 5 healthcare-11-02698-f005:**
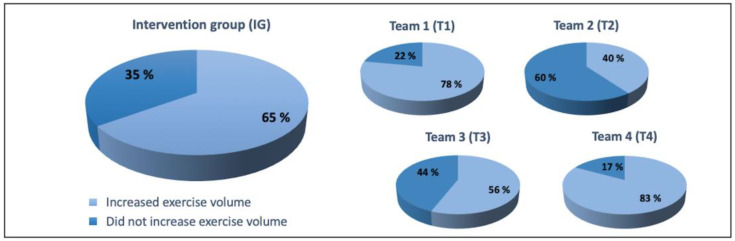
The reported subjective effect of the intervention to the exercise volume during the study period on a personal level for the IG. The participants reported if they increased exercise volume during the study period (12 months) due to the intervention when compared to the preceding 12 months.

**Table 1 healthcare-11-02698-t001:** The demographics of the participants. Mean and standard deviations (SD) are presented.

	CG (*n* = 32)	IG (*n* = 45)	T1 (*n* = 12)	T2 (*n* = 10)	T3 (*n* = 10)	T4 (*n* = 13)
Age (y)	31 (7)	34 (7)	36 (8)	34 (7)	34 (8)	34 (8)
Height (m)	1.80 (0.07)	′ 1.81 (0.06)	1.83 (0.07)	1.77 (0.06)	″ 1.80 (0.06)	1.82 (0.05)
Body mass (kg)	84.6 (12.3)	′ 86.8 (12.0)	89.5 (13.6)	84.3 (12.0)	″ 87.7 (15.5)	85.7 (8.0)
BMI (kg/m^2^)	26.1 (3.3)	′ 26.5 (3.1)	26.7 (3.6)	26.8 (3.2)	″ 26.9 (3.9)	25.9 (1.9)

Note. CG = control group, IG = intervention group, T1 = team 1, T2 = team 2, T3 = team 3, T4 = team 4, BMI = body mass index, ′ means *n* = 44, ″ means *n* = 9.

**Table 2 healthcare-11-02698-t002:** Mean (SD) reported weekly exercise volume during the intervention.

	IG	T1	T2	T3	T4
Endurance training (min)	42 (35)	32 (26)	36 (41)	37 (30)	64 (34)
Strength training (min)	30 (25)	15 (12)	13 (15)	33 (18)	60 (20)
Total time (min)	73 (53)	47 (32)	49 (50)	70 (42)	124 (44)

Note. IG = intervention group, T1 = team 1, T2 = team 2, T3 = team 3, T4 = team 4.

## Data Availability

The datasets used and analyzed during the current study are property of the Navy Command Finland. All data are primarily not public, but the data availability can be sought from the corresponding author on reasonable request.

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
