# Peer review of "Effects of 12-Month Training Intervention on Physical Fitness, Body Composition, and Health Markers in Finnish Navy Soldiers"

_healthcare, 2023, doi:10.3390/healthcare11192698_

Round 1
Reviewer 1 Report
Peer Review Report for: "Effects of a 12-Month Training Intervention on Physical Fitness, Body Composition, and Health Markers in Navy Soldiers"
For the journal: Healthcare MDPI
The article presents valuable findings; however, it necessitates significant revisions before it can be considered for publication.
1. Please specify the country in the study title to read: "in Finnish Navy Soldiers."
2. The background section of the abstract requires improvement. It is currently not well-articulated and needs to be rephrased. Can the authors provide evidence to support the claim that Navy soldiers are obese? Additionally, it's unclear which country's Navy soldiers are being referred to.
3. It would be beneficial to include specific numerical values in the abstract, such as p-values, Cohen's d, or odds ratios, consistent with the metrics used in your results section.
4. All keywords should be reformatted in the MeSH style. Ensure there are no duplicate terms from the title to enhance the visibility of your article.
5. Introduction: The flow of ideas in this section is disjointed. Kindly refine the structure to ensure a coherent presentation of ideas.
6. Methodology:
- The sampling size section appears to be missing. Please include this information.
- It would be helpful to provide a flowchart detailing the entire study protocol.
- More information is needed about the Navy soldiers involved in the study. Specifically, from which country are they, and do they belong to the same fleet or different ones?
7. The details of the training program throughout the intervention should either be illustrated in a graph or provided in-depth as supplementary files. I suggest converting section 2.2 into a graphical representation or elaborating on it in a supplementary file or additional table.
8. The statistical analysis is commendably executed.
9. Please incorporate Cohen's d to assess the level of improvement (significance).
10. Lastly, I urge the authors to adopt a more diplomatic and objective tone in their manuscript, especially given that it contains strategic information about their Navy soldiers.
Need improvement
Author Response
Responses to Reviewer 1 Comments
Dear Reviewer,
Thank you for your constructive comments. We have now revised our manuscript based on the reviewers' comments. We think the quality of the manuscript has improved from the previous version and hope that it now satisfies the editors and reviewers of the journal.
The article presents valuable findings; however, it necessitates significant revisions before it can be considered for publication.
- Please specify the country in the study title to read: "in Finnish Navy Soldiers."
Response: Thank you for your comment. This is specified and "in Finnish Navy Soldiers" has been added to the title.
- The background section of the abstract requires improvement. It is currently not well-articulated and needs to be rephrased. Can the authors provide evidence to support the claim that Navy soldiers are obese? Additionally, it's unclear which country's Navy soldiers are being referred to.
Response: Good remark. Only in the introduction section this is cleared out that in the U.S., military personnel aged over 20 years have been found to be more obese or overweight in the navy (67.0%) than in any the other military branch. There is no evidence that this is the same with all other nationalities. Therefore, that part is deleted from the abstract and the background section of the abstract is rephrased.
- It would be beneficial to include specific numerical values in the abstract, such as p-values, Cohen's d, or odds ratios, consistent with the metrics used in your results section.
Response: Thank you for your comment. We agree on this. Regarding to MDPI Healthcare's instruction for Authors, the abstract should be a total of about 200 words maximum. The present abstract is already 253 words and the results were so comprehensive that we therefore decided to exclude the p-values from the abstract.
- All keywords should be reformatted in the MeSH style. Ensure there are no duplicate terms from the title to enhance the visibility of your article.
Response: Thank you again for this idea. All keywords have now been reformatted in the MeSH style to enhance the visibility of the article.
- Introduction: The flow of ideas in this section is disjointed. Kindly refine the structure to ensure a coherent presentation of ideas.
Response: Thank you for your comment. The introduction section is now refined to create the flow of ideas more logical.
- Methodology:
- The sampling size section appears to be missing. Please include this information.
Response: Good remark. The reason behind the missing sampling size section is the fact that it was not possible to recruit more participants to the study from the fleets. Therefore, we did not calculate sampling size. The study was allowed to be performed only with two missile patrol boat fleets and in addition to research purposes, it was designed to increase the physical activity of the naval soldiers in practice.
- It would be helpful to provide a flowchart detailing the entire study protocol.
Response: Thank you for your comment. Figures 1 and 2 have been added to visualize the study protocol and measurements better.
- More information is needed about the Navy soldiers involved in the study. Specifically, from which country are they, and do they belong to the same fleet or different ones?
Response: Thank you again for a good comment. The nationality of the Navy soldiers has now been added as well as the Figure 1 to visualize that the participants are from two different missile patrol boat fleet.
- 7. The details of the training program throughout the intervention should either be illustrated in a graph or provided in-depth as supplementary files. I suggest converting section 2.2 into a graphical representation or elaborating on it in a supplementary file or additional table.
Response: Good remark. We agree on this. In the revised version the section 2.2 is modified so that the detailed information about the training sessions is converted to an additional table (Figure S2) that is located in the supplementary material.
- The statistical analysis is commendably executed.
Response: Thank you for the comment.
- 9. Please incorporate Cohen's d to assess the level of improvement (significance).
Response: Thank you for the suggestion to incorporate Cohen's d. We agree that proper estimates of effect size are important for assessing the findings, and Cohen's d provides a standardized measure of effect size. However, considering the expected readership of the present manuscript, we think that reporting the absolute differences between different timepoints, without standardization, allows more efficient communication of the results and lets the readers to assess the clinical significance of the intervention. Providing absolute differences is particularly important because the observed effects were rather small in most comparisons in the present study. We also think that reporting both the absolute differences and standardized effect sizes (i.e., Cohen's d) in the text, would markedly decrease the readability of the manuscript given the large number of comparisons included in the study. Therefore, we eventually came to the conclusion that providing only the absolute differences better serves the efficient communication of the results, and we have not added Cohen's d in the text. However, regarding the results of physical fitness tests, we have modified the supplementary Table S1 to show Cohen's d for each of the comparisons between IG and CG to serve the readers who wish to assess the findings in terms of standardized measures of effect size.
- Lastly, I urge the authors to adopt a more diplomatic and objective tone in their manuscript, especially given that it contains strategic information about their Navy soldiers.
Response: Thank you also for this valuable comment. The Surgeon General of the Finnish Defence Forces and the Commander of Coastal Fleet are aware of the present manuscript and it has been ensured that the information contained is allowed to be published in the form of the present manuscript. The information presented describe only the characteristics of the naval soldiers that participated in the study, not the FDF Navy soldiers' or the fleets in general.
Reviewer 2 Report
Thank you for submitting to Healthcare. I am honored to review.
The authors' document is well written and they have conducted meaningful research.
However, it is recommended that some additional information be included.
It is recommended that Table 2 and Figure 3 include information that can confirm statistical significance.
What does this research mean for the Navy? What is the difference compared to the Army and Air Force? Please compare broadly with previous studies.
I have no special opinion.
Author Response
Responses to Reviewer 2 Comments
Dear Reviewer,
Thank you for your kind words and constructive comments. We have now revised our manuscript based on the reviewers' comments. We think the quality of the manuscript has improved from the previous version and hope that it now satisfies the editors and reviewers of the journal.
Thank you for submitting to Healthcare. I am honored to review.
The authors' document is well written and they have conducted meaningful research.
However, it is recommended that some additional information be included.
It is recommended that Table 2 and Figure 3 include information that can confirm statistical significance.
Response: Thank you for your good remark and recommendation. We had a long consideration about this issue also when writing the original version of the manuscript.
Eventually we decided to keep the table and figure only on a descriptive level and did not add statistical tests because there were no results from the control group (CG). Considering all other results in the study, we compared the intervention group (IG) or the teams to the CG. In addition, the number of participants in a single team was quite low for reaching a statistical significance when comparing only the teams with one another.
Regarding the Figure, in hindsight, it would have been adequate to make a questionnaire also for the CG in which they would have been reported if they increased exercise volume during the study period (12 months) when compared to the preceding 12 months. Considering the Table, we did not have any reports of the amount of physical activity of the participants in the CG, while valid weekly reports were considered too challenging to obtain without any kind of intervention or motivation.
What does this research mean for the Navy? What is the difference compared to the Army and Air Force? Please compare broadly with previous studies.
Response: Thank you for your comment. When comparing to other military branches, the possible difference in the naval environment is the long-lasting stay in a confined space, especially when serving on a missile patrol boat. This might impair the endurance performance of the naval soldiers who work on small vessels and do not have the possibility to exercise on land during duty at sea. This issue has been added to the discussion section.
Otherwise, considering Navy, the present study supports earlier findings, which suggest that in a military environment carrying out a training intervention in a supervised manner may be more important than in a civilian environment. As mentioned in the discussion, this may be because the military is a controlled environment and works via orders, which could potentially inhibit the completion of uncontrolled voluntary duties. We think that there is still lack of knowledge on the voluntary physical activity interventions in the military environment (and even more in the Navy) to make additional further conclusions.
Reviewer 3 Report
First of all, I would like to thank the authors for the presented results of their investigation of navy solders. I especially appreciate such kind of effort bearing in mind how it is not easy to organize research on this population that last for one year. Also, I would like to thank the editor for the opportunity to review this manuscript.
The manuscript entitled "Effects of 12-month training intervention on physical fitness, body composition and health markers in navy soldiers " sought to examine the consequences of a year-long voluntary training program on navy sailors. In my opinion, the authors present an interesting topic that falls within the aims and scope of the Healthcare journal, special issue Health, Safety, and Readiness of Tactical Populations. The existing body of research on physical training interventions in military settings has been focused on short-term interventions that are monitored. However, the effectiveness of these interventions in producing long-term voluntary training benefits has been found to be insufficient.
The work has a good outline and is easy to read, although there is room for improvement. My main concern is relating to more precision and coherence in some parts of the manuscript. In this form, there is an imbalance between the Introduction and other parts, dominated by the emphasis on motivation. In this sense, the Introduction should contain a part related to motivation (e.g., intrinsic and extrinsic). Further, the first part of the title should more precisely indicate the effect generators (coaches). Moreover, in parallel with the level of motivation of the participants observed by the trainers, it could be assessed through a questionnaire. An intervention related to the above could improve the manuscript.
I encourage the authors to consider options that it is not strictly necessary to develop capabilities in the tactical population, e.g., physical, but that it is also possible to maintain the achieved level if it is satisfactory. This is also connected with the fact that members of tactical populations have already been selected and should reach a certain level of ability through courses. Finally, increasing the exercise volume may be desirable, but quality and content based on desired outcomes are more important.
Lines 58-60: Consider linking words to avoid citing the same reference in two consecutive sentences.
Lines 71-75: Consider linking words to avoid citing the same reference in two consecutive sentences.
Lines 90, 91: It would be correct to say that each IG comprised participants of the same vessel.
Lines 91, 92: What was rationale for including female only in CG?
Lines 92, 93: This is redundant as it was already mentioned in the Table 1.
Lines 120, 121: How did the coaches motivate their team?
Lines 123-125: What were the motivation elements in the weekly email?
Line 326, Table 2: One minute is missing in IG.
Lines 375-377: It is also worth emphasizing the positive influence on body composition parameters. Although it is a statistically negative correlation, it is positive in the essential meaning.
Best wishes
Author Response
Responses to Reviewer 3 Comments
Dear Reviewer,
Thank you for your kind words, appreciation and constructive comments. The 12 months of work with this intervention was a great effort indeed. We have now revised our manuscript based on the reviewers' comments. We think the quality of the manuscript has improved from the previous version and hope that it now satisfies the editors and reviewers of the journal.
First of all, I would like to thank the authors for the presented results of their investigation of navy solders. I especially appreciate such kind of effort bearing in mind how it is not easy to organize research on this population that last for one year. Also, I would like to thank the editor for the opportunity to review this manuscript.
The manuscript entitled "Effects of 12-month training intervention on physical fitness, body composition and health markers in navy soldiers " sought to examine the consequences of a year-long voluntary training program on navy sailors. In my opinion, the authors present an interesting topic that falls within the aims and scope of the Healthcare journal, special issue Health, Safety, and Readiness of Tactical Populations. The existing body of research on physical training interventions in military settings has been focused on short-term interventions that are monitored. However, the effectiveness of these interventions in producing long-term voluntary training benefits has been found to be insufficient.
The work has a good outline and is easy to read, although there is room for improvement. My main concern is relating to more precision and coherence in some parts of the manuscript. In this form, there is an imbalance between the Introduction and other parts, dominated by the emphasis on motivation. In this sense, the Introduction should contain a part related to motivation (e.g., intrinsic and extrinsic). Further, the first part of the title should more precisely indicate the effect generators (coaches). Moreover, in parallel with the level of motivation of the participants observed by the trainers, it could be assessed through a questionnaire. An intervention related to the above could improve the manuscript.
Response: Thank you for a good comment. To correct this imbalance between the introduction and other parts, information on motivational physical activity interventions has been added to the introduction section.
I encourage the authors to consider options that it is not strictly necessary to develop capabilities in the tactical population, e.g., physical, but that it is also possible to maintain the achieved level if it is satisfactory. This is also connected with the fact that members of tactical populations have already been selected and should reach a certain level of ability through courses. Finally, increasing the exercise volume may be desirable, but quality and content based on desired outcomes are more important.
Response: Thank you again for a good comment. It has been added to the discussion section that naval soldiers have a strict selection process, requirements for the minimum level of physical fitness, and soldiers in overall have been found to be more physically active than civilians. Therefore, it may not always be strictly necessary to enhance soldiers' physical fitness if they have maintained it on a satisfactory level.
Lines 58-60: Consider linking words to avoid citing the same reference in two consecutive sentences.
Response: Good remark. This is now corrected.
Lines 71-75: Consider linking words to avoid citing the same reference in two consecutive sentences.
Response: Again, a good remark. This is now corrected.
Lines 90, 91: It would be correct to say that each IG comprised participants of the same vessel.
Response: Thank you for your comment. This is now corrected.
Lines 91, 92: What was rationale for including female only in CG?
Response: Good question. The study was performed with two missile patrol boat fleets and in addition to research purposes, it was designed to increase the physical activity of the fleets' naval soldiers in practice. In CG's fleet there just happened to be two female and because of the practical aspect they were not excluded from the study. Solely on research purposes only male participants would have been more explicit option. On the other hand, in the present study we only studied the personal differences in physical fitness tests, body composition and health makers, so the gender did not have a considerable impact.
Lines 92, 93: This is redundant as it was already mentioned in the Table 1.
Response: Thank you again for your comment. This sentence is now deleted from the manuscript.
Lines 120, 121: How did the coaches motivate their team?
Response: Good question. It has now been added to the manuscript that the coaches organized training sessions and verbally motivated their team members to carry out the weekly training.
Lines 123-125: What were the motivation elements in the weekly email?
Response: Again, a good question. One motivation element of the weekly email was of course to remind the participants that the intervention was going on during the long 12-month study period. In the e-mail, depending on the reported amounts of weekly training, the main researcher either thanked the IG for a good work or encouraged to add training volume. This information has now been added to the revised manuscript.
Line 326, Table 2: One minute is missing in IG.
Response: Good remark. This is due to the rounding of the values.
Lines 375-377: It is also worth emphasizing the positive influence on body composition parameters. Although it is a statistically negative correlation, it is positive in the essential meaning.
Response: Again, a good remark. This has been added to the revised manuscript.
Round 2
Reviewer 3 Report
Dear authors,
Thank you for your time and effort in responding to the comments. The manuscript is now improved.
Best wishes.